# Schistosomiasis with a Focus on Africa

**DOI:** 10.3390/tropicalmed6030109

**Published:** 2021-06-22

**Authors:** Oyime Poise Aula, Donald P. McManus, Malcolm K. Jones, Catherine A. Gordon

**Affiliations:** 1School of Public Health, Faculty of Medicine, University of Queensland, Brisbane 4006, Australia; Don.McManus@qimrberghofer.edu.au; 2Molecular Parasitology Laboratory, QIMR Berghofer Medical Research Institute, Brisbane 4006, Australia; 3School of Veterinary Sciences, University of Queensland, Gatton 4343, Australia; m.jones@uq.edu.au

**Keywords:** schistosomiasis, *Schistosoma haematobium*, *Schistosoma mansoni*, sub-Saharan Africa, Africa

## Abstract

Schistosomiasis is a common neglected tropical disease of impoverished people and livestock in many developing countries in tropical Africa, the Middle East, Asia, and Latin America. Substantial progress has been made in controlling schistosomiasis in some African countries, but the disease still prevails in most parts of sub-Saharan Africa with an estimated 800 million people at risk of infection. Current control strategies rely primarily on treatment with praziquantel, as no vaccine is available; however, treatment alone does not prevent reinfection. There has been emphasis on the use of integrated approaches in the control and elimination of the disease in recent years with the development of health infrastructure and health education. However, there is a need to evaluate the present status of African schistosomiasis, primarily caused by *Schistosoma mansoni* and *S. haematobium*, and the factors affecting the disease as the basis for developing more effective control and elimination strategies in the future. This review provides an historical perspective of schistosomiasis in Africa and discusses the current status of control efforts in those countries where the disease is endemic.

## 1. Introduction

Schistosomiasis is a snail borne, fresh water-transmitted neglected tropical disease (NTD) of poverty in many developing countries in tropical and sub-tropical Africa, the Middle East, Asia, and Latin America [1,2,3] (Table 1). The disease is generally endemic in low-income rural communities without access to potable water, proper sanitation, and adequate healthcare facilities. Sub-Saharan Africa (SSA) constitutes about 13% of the world’s population but accounts for up to 90% of cases with an estimated 280,000 deaths due to schistosomiasis annually [2]. The two major species infecting humans in sub-Saharan Africa are *Schistosoma haematobium*, which causes urogenital schistosomiasis and *S. mansoni*, the cause of intestinal schistosomiasis. *S. intercalatum* and *S. guineensis* also cause intestinal schistosomiasis but are less prevalent (Table 1) [4]. Similarly, the main schistosome species infecting animals are *S. mattheei, S. bovis* and *S. currassoni* (Table 1). In recent years, there have been cases of locally acquired urinary schistosomiasis cases caused by *S. haematobium* reported in Corsica, France where it had previously not been endemic, likely imported by individuals infected in West Africa [5,6]. There is also increasing evidence of hybridization events between human and animal schistosome species, leading to new zoonotic infections [7,8,9].

Schistosomiasis control programs in Africa focus predominantly on community-based preventive chemotherapy (PC), focusing on mass drug administration (MDA) with praziquantel (PZQ), a broad spectrum anthelminthic, to reduce morbidity. Every year PZQ is donated to endemic countries in Africa to treat millions of school-aged children (SAC) [16,17]. Treatment compliance is challenging, especially among individuals in low socioeconomic areas due to fear of adverse effects, apparent absence of disease symptoms, and even when symptoms occur, they are often met with stigmatisation [18,19] or seen as a normal sign of puberty [20,21] not requiring treatment. There have been increased efforts aimed at eliminating schistosomiasis in the past decade with the World Health Organization (WHO) setting 2030 as the goal for transmission interruption in endemic African countries [22]. Government agencies from many countries have prioritised the control of NTDs by exploiting breakpoints in their lifecycles, such as the implementation of snail control and through improvements in sanitation and access to safe, clean water [23,24]. However, in sub-Saharan Africa, PC is still the major intervention practiced. As PZQ is ineffective against migrating schistosomula, if these larvae are present in an individual, treatment may not prevent their maturation to adults and resultant patency. Rapid re-infection following treatment with PZQ is also common [25,26,27].

SAC are the main targets for control as they are considered to be at the highest risk of infection, being more likely to participate in daily activities including fishing, rice farming and swimming, that put them at greater risk of infection compared with other age groups. Protective immune responses against schistosomes develop slowly, with children from schistosomiasis-endemic areas being generally susceptible to reinfection after treatment for schistosome infection, whereas adults are usually protected [10]. This observation can explain the characteristic age-prevalence and age-intensity curves observed in schistosomiasis-endemic areas in Africa [11,28,29].

## 2. Pathogenesis

### 2.1. Life Cycle of Schistosoma sp.

Schistosomes have a complex life cycle involving both intermediate gastropod hosts and a definitive mammalian host (Figure 1). Unlike other trematode species, *Schistosoma* spp. are dioecious (separate male and female worms) which undergo sexual reproduction in the mammalian definitive host. Schistosome eggs are produced and excreted into the environement via the faeces (*S. mansoni*) or urine (*S. haematobium*). Miracidia are released when the eggs come in contact with water and infect the snail host. There, miracidia develops into mother sporocysts and undergo asexual reproduction to produce daughter sporocysts which produce cecariae. Infected snails shed cercariae into the water and upon locating a suitable definitive host, penetrate the skin, transform into schistosomula and migrate through the circulatory system to the lungs, heart and liver where they mature into adult worms (Figure 1). The adult worms then exit the liver and pair up to mate in the mesenteric vessels of the bowel (*S. mansoni, S. intercalatum*) or bladder (*S. haematobium*).

A proportion of the eggs are carried by the bloodstream to other areas of the body where they can become lodged in tissues, and trigger an inflammatory response, causing acute or chronic disease. (Figure 1). Schistosomes have an average life span of 3–10 years but can live up to 40 years in their human hosts in permanent copulation [30,31].

The control and elimination of schistosomiasis requires interruption of a complex pathway of transmission governed by the interplay of humans, intermediate host snails and human–water contact patterns. The snail hosts are crucial for determining the range of schistosomiasis and are responsible for the focal nature of the disease (i.e., highly variable infection prevalence and intensity even within a small area such as from one village to another). Two genera of snails (*Bulinus* and *Biomphalaria*) are responsible for the distribution of schistosomes in Africa (Table 1). These molluscs can be an important focus of control efforts involving environmental modification (e.g., digging water drainage ditches or tunnels to flood and bury the snail habitats to disrupt snail habitats), or through the use of chemicals, such as niclosamide [32,33]. Concerning, however, are the detrimental impacts that such chemicals can have on the environment including the general pollution they cause and being toxic to larger animals such as fish [34,35].

### 2.2. Clinical Presentation of Schistosomiasis in Africa

Schistosome infection has three distinct disease phases beginning with an initial dermatitis reaction following skin penetration by the cercariae resulting in an allergic inflammatory maculopapular lesion [36]. The infection may then proceed to a symptomatic acute schistosomiasis stage also known as Katayama fever or Katayama syndrome. Acute schistosomiasis is rarely reported in individuals living in areas endemic for *S. mansoni* or *S. haematobium*. One possible explanation for this being that in-utero sensitisation might decrease the severity of common symptoms of Katayama syndrome in chronically exposed individuals resulting in lowered immune responsiveness to schistosome antigens in infants born to infected mothers; it may also be equally likely that cases from endemic areas are simply unrecognised or under-reported [36,37]. The most common symptoms of acute schistosomiasis include prolonged fever, weakness, vomiting, nausea, diarrhoea, malaise and rapid weight loss [4,38]. The third and final disease stage, chronic schistosomiasis, occurs when eggs are deposited in various body tissues, commonly affecting the liver, bladder and urogenital system, and less commonly in the central nervous system [4,39]. Adult worms can avoid detection by the immune system by camouflaging their outer layer with host antigens and tegmental shedding and are able to reside for long periods in their hosts [40]. In contrast, schistosome eggs are fully exposed to the immune system, and this results in the formation of granulomatous and fibrotic lesions around the eggs in various tissues resulting in necrosis, ulceration and bleeding that can have long-term detrimental effects [4,38,41,42]. Chronic schistosomiasis is associated with hepatosplenomegaly, portal fibrosis and, in the case of *S. haematobium*, haematuria (blood in urine), ureter fibrosis, and squamous cell carcinoma of the urinary bladder [4].

*S. mansoni* is the leading cause of intestinal schistosomiasis in Africa (Table 1). Around 50% of eggs deposited by adult worms are retained in the liver, causing chronic disease [43]. Pathogenesis due to *S. intercalatum* is less severe than *S. mansoni* and *S. haematobium* and most infected patients, particularly children, do not show symptoms of the disease [44].

#### 2.2.1. Female Genital Schistosomiasis

Female genital schistosomiasis (FGS) is characterised by the presence of *S. haematobium* eggs in the vagina and cervix and affects up to 20 million women in sub-Saharan Africa and the Middle East [45,46]. The eggs penetrate the urogenital system, causing uterine enlargement, menstrual disorders, cervicitis and infertility [47]. Schistosomiasis in pregnant women presents with symptoms ranging from anaemia during pregnancy to newborns with low birth weight, and increased infant and maternal mortality rates [48,49,50,51]. Urogenital schistosomiasis has also been linked with increased risk of HIV infection in women resulting from the production of genital mucosal lesions surrounding the eggs [52,53]. The immune response caused by *S. haematobium* infection leads to chronic inflammatory granulomatous lesions, genital epithelial bleeding and sandy patches in the cervix and vagina that, if left untreated, can become an easy entry point for the HIV virus, as well as leading to infertility [30,52,54,55,56,57]. Concurrent infection with HIV and *S. haematobium* leads to increased disease pathology while HIV infection may lead to an increased chance of contracting schistosomiasis [52,58,59]. Schistosomiasis has also been suspected of increasing disease progression and death in HIV patients by increasing the HIV RNA load in the blood plasma [60,61]. More than 70% of HIV infections worldwide occur in sub-Saharan Africa and thus HIV remains a major health challenge in Africa and an important confounding factor for schistosome infection. Diagnosis of FGS can be challenging due to different transformation stages of the *S. haematobium* parasite and immune response in the affected tissues. In cases where the infected patient is asymptomatic, the disease may be mistaken for sexually transmitted diseases (STDs) or cervical cancer [18,46,62]. Stigma against STDs can lead to misdiagnosis, or reluctance of young women to present to a doctor when experiencing clinical symptoms of FGS.

#### 2.2.2. Primary and Secondary Infertility in *S. haematobium* Infections

Infertility is the inability to become pregnant after regular and unprotected sexual intercourse for more than one year [63]. It can be diagnosed as either primary–where the woman has never conceived, or secondary-when the woman has experienced previous labour. Suspected cases of infertility resulting from *S. haematobium* infection in the genital tracts have been widely reported in Africa [64,65,66,67,68,69,70]. While the presence of an adult worm infection in the urogenital system is generally asymptomatic, the deposition of *S. haematobium* eggs along the urogenital tract, including the cervix and vagina, trigger a hypersensitive immune response, causing scarring and fibrosis in the genital tract, ovaries and fallopian tubes. The eggs may appear as papillary white lesions [64,65], causing thickening, nodular lesions and adhesions which eventually lead to obstruction and blockage of the fallopian tubes. The resulting fibrosis and blockage is suspected to lead to infertility. A case study in Nigeria reported the inability of a woman to get pregnant with her second child despite having a regular menstrual cycle; tuboplasty (surgery undertaken to restore the functionality and integrity of the fallopian tubes) revealed lesions and blockage of the patient’s fallopian tubes due to the presence of *S. haematobium* eggs [65]. Urogenital schistosomiasis has also been linked to ectopic pregnancies as a result of blockage of the fallopian tubes [66,67]. Patients can recover with administration of PZQ if the infection is treated sufficiently early [65,66].

#### 2.2.3. Male Genital Schistosomiasis

Male genital schistosomiasis (MGS) was first reported in 1911 [71] and is described as the presence of schistosome (*S. haematobium*) eggs in the male genital organs and fluids. The awareness and severity of this disease especially in endemic areas is often overlooked and underreported as it can be misdiagnosed as a sexually transmitted infection (STI) [59,72,73]. Symptoms of this disease include painful urination, painful ejaculation, irregular ejaculations, hermatospermia, prostatitis, epididymitis (inflammation of the epididymitis at the back of the testicles), which could mimic tuberculosis and associated funiculitis, erectile dysfunction, enlarged genital organs and infertility [59,74,75,76].

#### 2.2.4. Bladder Cancer in *S. haematobium* Infections

Globally, 275,000 people are diagnosed with bladder cancer annually and 108,000 people die of the disease. Bladder cancer caused by translational cell carcinoma (TCC) occurs in industrialised and developing countries not endemic for urogenital schistosomiasis, while bladder cancer caused by squamous cell carcinoma (SCC) is a long-term sequela of chronic infection and occurs in many parts of Africa plagued with urogenital schistosomiasis [77,78]. Bladder cancer is one of the foremost serious complications of chronic *S. haematobium* infection and it is estimated that the schistosome-associated bladder cancer incidence is 3–4 cases per 100,000 infections [79].

TCC arises from the transitional epithelium lining of the bladder and presents in its early stage as a painless haematuria. In contrast, a squamous hyperplasia (usually not present in a normal urothelium which is a highly specialized epithelium lining the lower urinary tract) gives rise to SCC due to injuries caused by the immunological responses to deposited eggs in the bladder [80]. This is followed by painful haematuria, chronic inflammation and necroturia [80]. SCC often presents with symptoms only at a late stage and can be challenging to treat by surgery or with chemotherapy.

A study in Egypt reported that 82% of patients with SCC had *S. haematobium* eggs lodged in their bladder wall and infected individuals had the tendency to develop cancer at a younger age than uninfected individuals [81]. Kitinya et al. [82] reported that of 172 individuals with bladder cancer in Egypt over a nine year period (1971–1980), 72% were SCC cases. Similarly, a study in Northern Tanzania reported 46% of SCC patients had *S. haematobium* eggs in the tumour tissues [82]. Another study from Angola, situated in the western part of Western Africa, reported a >70% (215/300) *S. haematobium* prevalence with 3 of the infected patients having calcified bladders and one SCC case was recorded [83]. A decrease in the prevalence of urogenital schistosomiasis in Egypt has seen a decline in the SCC and an increase in the median age of infected individuals with bladder cancer [84].

The mechanisms associated with *S. haematobium* and the development of SCC are largely unknown although the carcinogenic process appears to be closely related to tissue inflammation [3]. Botelho et al. [85] described the relationship between *S. haematobium* and cancer by exposing normal epithelial cells (Chinese Hamster Ovary (CHO) cells) in culture with *S. haematobium* total antigen and observed increased cell proliferation, decreased apoptosis, migration, invasion and tumourigenesis [85,86]. This suggested that *S. haematobium* has the ability to induce the formation of cancer-like cells [85]. Furthermore, *S. haematobium* exposed cells injected into mice with no immune system resulted in the development of tumours similar to those found in bladder cancer [87].

### 2.3. Treatment and Control

Preventive chemotherapy (PC) through MDA with PZQ is the cornerstone of the treatment and control of schistosomiasis in endemic regions of Africa. PZQ has been proven a safe and effective oral drug active against adult worms of all *Schistosoma* species [88,89], although its mechanism of action is still not fully understood. However, it cannot be used for chemoprophylaxis due to its short half-life, and it is ineffective against migrating schistosomula [90]. Corticosteroids, in addition to PZQ, are effective as an adjuvant when patients present with Katayama syndrome, usually within two months of exposure to cercariae [37,91] to suppress immunological reactions and prevent acute disease. Other drugs that proved effective for the treatment of schistosomiasis include oxamniquine for *S. mansoni,* and metrifonate for *S. haematobium* [4,38] but these are either no longer readily available or have been withdrawn due to unacceptable toxicity. Co-infections of *Schistosoma* spp. and soil-transmitted helminths (STH) are common in many endemic areas in Africa, and as such, combination PC with both PZQ and albendazole is recommended by WHO [92] particularly for SAC and other high risk groups.

PZQ is given to SAC between the ages of 5 and 15 years who have the highest infection rates and are more readily reached through school programs. PC is usually carried out by firstly assessing the prevalence of the disease which determines the frequency of treatment in that area [93]. For example, areas showing disease prevalence with 50% or more usually should receive a single annual treatment while areas with 10% prevalence will receive triennial treatment [93]. As of 2019, 57.1% (61.8 million) SAC who require treatment have received PZQ [94].

Re-infection remains a major challenge to control efforts in Africa due to a number of factors including: high levels of infection prevalence and intensity, poor or non-compliance of PZQ treatment and low coverage, recontacting contaminated water as a result of daily activities and seasonal factors. Hence, a multifaceted intervention approach will be needed to move from wide-spread control to elimination including: snail control; treatment; effective risk mapping and epidemiological surveillance; accurate diagnostics; improved access to clean water, sanitation and hygiene (WASH); and public health education to bring about behavioural changes to prevent infection and reinfection [95,96,97,98,99]. These integrated approaches, together with the development and deployment of future anti-schistosome vaccines effective in humans (albeit no schistosomiasis vaccine has yet been accepted for public use) will contribute greatly to reducing and interrupting transmission in endemic areas leading to eventual elimination [100,101]. Another challenge to be faced is climate change and the resultant elevated temperatures which may increase the geographical distribution of the parasite through expansion of suitable environments for snails into higher altitudes and into further locations in Africa currently unaffected by the disease. While most studies focus on increasing temperature, it has been shown that snails and schistosomes within their hosts survive during the winter months and produce viable cercariae that complete their life cycles when optimal temperature is reached [102]. Furthermore, snails from temperate region demonstrate better resistance to harsh winter conditions than tropical snails [102].

### 2.4. Diagnosis

There are a number of approaches used for schistosomiasis diagnosis and schistosome detection. The standard method used in Africa is the detection of eggs in urine or stool by microscopy [103,104], although a number of immunological [105,106,107,108,109,110,111] and molecular [112,113,114,115,116,117,118,119] diagnostic assays have been developed with some deployed in Africa (Table 2). Polymerase chain reaction (PCR) and quantitative PCR (qPCR)-based molecular methods are now increasingly being employed for diagnosis in high-resource settings globally but they are expensive, take time, require a significant laboratory infrastructure and training which hampers their current use in low socio-economic endemic field settings. Isothermal amplification detection (IAD) methods can overcome some of these obstacles including the limitations of costly thermal cyclers required for the PCR-based detection of parasite DNA in stool or urine. IAD assays work similar to that of conventional PCR in that they utilise DNA or RNA polymerase in the extension of target-specific primers. However, isothermal amplification facilitates amplification without the repeated cycles of denaturation and annealing required for PCR. The most established IAD method for schistosome detection is loop-mediated isothermal amplification (LAMP) [120,121,122,123,124,125,126,127,128,129]. Other isothermal methods for parasite detection include helicase-dependent isothermal amplification (HDA) [130], recombinase polymerase amplification (RPA) [131] and nucleic acid sequence-based amplification (NASBA) [132], but only RPA has been applied in the detection of *Schistosoma* spp. [133,134,135,136].

As indicated earlier, the microscopic detection of eggs in urine (*S. haematobium*) and faeces (*S. mansoni*) is the most commonly used method for the diagnosis of schistosomiasis. The Kato-Katz (KK) method is used to detect *S. mansoni* eggs in faeces, while urine microscopy, preceeded by urine filtration, is used to identify *S. haematobium* infections [103,104]. *S. haematobium* eggs were identified in the semen of fishermen as part of a cross-sectional study along the southwestern shoreline of Lake Malawi in Sub-Saharan Africa, suggestive of high lodgement of eggs in the reproductive organs of men [76]. The precise origin of eggs found in semen is unresolved but they may have originated in the bladder, carried with drops of urine through the urethra and released with the semen [59]. Eggs of *S. haematobium* can be readily detected by light microscopy, and cell-free circulating schistosome DNA has been detected in semen several weeks after a single dose of PZQ [137].

In addition to egg detection, active infections can be detected from worm-derived circulating anodic antigens (CAAs) and circulating cathodic antigens (CCAs) in serum and urine using enzyme-linked immunosorbent assay (ELISA) or monoclonal- antibody-based lateral flow tests [138,139]. These detection methods have the ability to detect infection before the worms begin producing eggs, [140,141,142]. However, they do not discriminate between past, active or re-infections, especially in endemic areas where patients can remain seropositive several years after treatment [140,143].

Haematuria and proteinuria reagent strip testing can also be used as indirect diagnostic methods for *S. haematobium* infection [144]. Strip testing has previously been shown to provide sensitivity and specificity levels of 75% and 87%, respectively, for detection of *S. haematobium* [145] and has been suggested as an alternative form of diagnostic to the usual urine microscopy. Strip testing may be useful in sub-Saharan Africa due to its substantially higher sensitivity than microscopic methods and ease of storage of the strips [138]. However, the strip tests also detect haematuria not associated with *S. haematobium* infection, and the method exhibits poor sensitivity in detecting egg-positive urine post-treatment and low-intensity infections [138].

#### Environmental Monitoring

As indicated earlier, schistosomiasis is a highly focal disease with transmission being highly dependent on the presence of fresh water and appropriate snail intermediate hosts, as well as water contact activities by humans who become infected. The risk of infection is dependent on seasonal changes in snail populations, water levels, infection rates and cercarial output. Flooding events may also cause temporarily higher rates of infection in human communities. Information on snail hosts and the distribution of cercariae are important tools in the control and elimination of schistosomiasis [3,135,146].

Molecular xenomonitoring is a useful disease surveillance tool for the detection of infection rates in field population of snails and could be useful in identifying infection risk areas to help guide intervention measures for schistosomiasis control and elimination [147,148,149,150]. There are a number of methods used for xenomonitoring including sentinel mice which have been used to identify transmission sites in natural water bodies in China; however, this process is time-consuming and expensive [148,151]. Morphological identification of miracidia and cercariae collected from water sources can be inaccurate due to disintegration of the larvae and misidentification of human and non-human cercariae that co-exist in most endemic areas; the latter issue is also applicable to identifying cercariae from infected snails [152] (Figure 1, Table 1).

PCR-based detection methods have been developed that detect cercariae in water samples and schistosome species in snail intermediate snail hosts, and these have proved useful in identifying and monitoring schistosomiasis transmission areas in Africa [148,153]. An example is the DraI PCR, which has been used to monitor snail transmission *S. haematobium* cercariae in Morocco [154]. The DraI ribosomal sequence is specific to the *S. haematobium* group and can detect low amounts of DNA due to its abundant sequences in the *S. haematobium* genome [155,156,157,158]. Another example is a two-step multiplex PCR approach which first identifies *Schistosoma* infected snails, followed by species-specific identification using internal transcribed spacer (ITS) rRNA primers [159,160].

## 3. A Brief History of Schistosomiasis in Africa

Schistosomiasis in Africa dates back more than 4000 years [73,161,162,163]. Symptoms characteristic of urinary schistosomaisis were first described in early Egyptian papyri and the eggs of *S. haematobium* were identified in the urinary tracts of mummies from 1250–1000 BC [73,161,162,163]. Symptoms were described as urological problems such as enlarged prostate, bladder stones, cystitis, changes in urinary frequency and discharge from the penis; this has been interpreted as blood although it could have also indicated semen, or purulent discharge from sexually transmitted diseases [164,165]. More recently, there were reports of persistent haematuria recorded by members of Napoleon’s army in Egypt in 1798, and in forces involved in the Boer war (1899–1902) [164]. Schistosomiasis was first recorded in the Eastern Cape of South Africa in 1863, after Dr J Harley diagnosed endemic haematuria with unknown cause in local residents. He later diagnosed this as Bilharziasis after observing eggs in the urine 11 years after the official identification of the parasite [165,166,167,168]. It was described as common in the Cape and thought to be due to contact with freshwater [166,167]. Cases of haematuria later identified as schistosomiasis based on the presenting symptoms were wide-spread in South Africa between 1864 and 1899 with children particularly affected [166,169,170,171]. Women and girls were considered to be less affected as they had little contact with ‘natural’ water [166]. The geographical distribution of urogenital schistosomiasis in South Africa was later described and efforts were made to recognise the disease as a serious public health issue from the 1960s onwards [169,170,171]. However, government run-control programmes were essentially non-existent until the 1990s when the first helminth control program, targeting soil-transmitted helminths as well as *S. haematobium,* was set up in the KwaZulu-Natal province of South Africa in 1997 [172].

The German physician Theodore Bilharz first officially identified the parasite causing schistosomiasis in 1851, after he recovered two distinct species from autopsies of dead soldiers in Egypt [165,167]. Dr Bilharz first named the parasite *Distomum haematobium* and also described hatching of eggs, linking the existence of the parasite to clinical symptoms–primarily haematuria–attributed to the disease [165,167]. Bilharzia was later adopted as the generic term for the schistosome parasites. In 1902, Sir Patrick Manson posited that humans could be infected with *S. haematobium* and another species of *Schistosoma* based on differences in egg morphology and the manner of excretion. This was not, however, officially accepted until 1915 when *S. mansoni* was established as a separate species distinct from *S. haematobium* by Louis Westenra Sambon who named it in honour of Sir Patrick Manson [173].

### 3.1. Current Status of Schistosomiasis in Africa

Schistosomiasis is a major ongoing public health issue particularly in sub-Saharan African countries [174] (Figure 2). *S. haematobium* is the most prevalent species in sub-Saharan Africa (Table 1) with an estimated 112 million individuals infected [175]. Nearly 71 million individuals experience haematuria, half of which have dysuria, and around 18 million infected individuals suffer from urinary bladder pathology annually [175]. Deaths resulting from kidney failure due to schistosomiasis haematobia amount to 150,000 annually [2,175]. There is an estimated 54 million individuals infected with *S. mansoni* with around 4 million people experiencing diarrhea and 8.5 million hepatomegaly [175], with deaths resulting from haematemesis estimated to be around 130,000 annually [2,175].

*S. haematobium* and *S. mansoni* are geographically present together in much of Africa (Figure 2) often leading to co-infections with both parasites [175] (Table 2). Nigeria in West Africa currently has the highest schistosomiasis prevalence in the world [178], with *S. mansoni* and *S. haematobium* recorded there since 1881 [179]. Urogenital schistosomiasis is widespread throughout the country while *S. mansoni* is both less prevalent and has a reduced geographical distribution [180,181]. Schistosomiasis is thought to have been transmitted to Northern Nigeria by Fulani herdsmen from the Upper Nile Valley, and the distribution of the infection was first mapped out in 1963 [179,181]. A World Bank report in 1997 estimated that 25 million individuals in Nigeria were infected with *S. haematobium*, *S. mansoni* or with both species [182]. As of 2015, 29 million people were estimated infected [183]. Prevalence studies have continued, however more recent figures of infection are not available despite this [180]. *S. intercalatum* is also present in Nigeria [184]. A survey in Nigeria examined 47 dams in areas hyper-endemic or moderately endemic for schistosomiasis. Of these dams, 20 had the requisite intermediate molluscan hosts, indicating a need for new large scale surveys to determine the true prevalence and distribution of schistosomiasis in Nigeria [180]. Recently, a mapping project for the distribution of urogenital schistosomiasis in Anambra state, Nigeria, was undertaken [185]. Participants (*n* = 450) in the area were recruited and asked to provide a urine sample which was tested using a dipstick to identify those with haematuria and egg microscopy to gauge prevalence in the study area. Overall, the prevalence was low with 5.5% infection in individuals diagnosed by haematuria, and 2.9% by microscopic detection of eggs [185], although both tests are considered to have low sensitivity so the true prevalence was likely underestimated [180,186]. In Katsina State, Nigeria, the prevalence of urogenital schistosomiasis reached 22.7% in 2018 and was higher than the prevalence (8.7%) determined in a separate study on SAC in the same area in 2016 [187,188] (Table 2). A very high prevalence (38.9%) was determined in Imo state in Oguta, although this report is nearly 20 years old [189]. Thus, reports of schistosomiasis in Nigeria are scattered, with limited recent studies and differing methodology used making it difficult to compare studies and to accurately estimate true overall country prevalence (Table 2).

**Table 2 tropicalmed-06-00109-t002:** The prevalence of schistosomiasis in different African countries. KK—Kato-Katz. qPCR—real-time polymerase chain reaction. POC_CCA—point-of-care circulating cathodic antigen. DDIA—dipstick dye immunoassay. IHA—indirect hemagglutination assay. Ss/Sp—sensitivity/specificity.

Country	Method	Species	Prevalence % (n/tn)	Ss/sp (%)	Study Type	Age (years)	Intensity of Infection (%)			Study Year	Study Published	Ref
							Light	Mode rate	High			
**Angola**	Urine microscopy	*S. haematobium*	61.18 (785/1283)	-	Cross-sectional survey	9–10	-	-	-	2013–2014	2015	[190]
	Urine dipstick	*S. haematobium*	65.8 (844/1283)	96/61			-	-	-			
	Haematuria	*S. haematobium*	17.1 (219/1283)	27.1/97.5			-	-	-			
	LAMP	*S. haematobium*	73.8 (127/172)	-	Evaluation	5–14	-	-	-	2015	2018	[191]
**Benin**	KK	*S. mansoni*	2.45 (472/19250)	-	Surveillance	8–14	59.32	25.42	15.25	2013–2015	2019	[192]
	Urine microscopy	*S. haematobium*	17.60 (3388/19250)	-			73.99	-	20.01			
**Burkina Faso**	KK	*S. mansoni*	5.38 (43/800)	-	Prevalence	7–11	-	-	-	2013	2016	[193]
	Urine microscopy	*S. haematobium*	8.76 (287/3514)	-			-	-	2.7			
**Cameroon**	KK	*S. mansoni*	61 (381/625)	-	Evaluation	7–15	-	-	-	2010–2011	2012	[110]
	Urine-CCA	*S. mansoni*	66.6 (416/625)	-			-	-	-			
	Urine microscopy	*S. haematobium*	4.6 (29/625)	-			-	-	-			
	Dipstick	*S. haematobium*	9.8 (61/625)	-			-	-	-			
**Chad**	Urine microscopy	*S. haematobium*	24.9 (467/1875)	-	Prevalence	1–14	-	-	-	2015–2016	2019	[194]
**Côte d’Ivoire**	Urine microscopy	*S. haematobium*	14 (166/1187)	-	Cross-sectional survey	5–14	-	-	-	2018	2019	[195]
	KK	*S. mansoni*	6.1 (66/1089)	-			-	-	-			
	CCA	*S. mansoni*	73.8 (104/141)	-	Cross-sectional survey	8–12	-	-	-	2010	2011	[109]
	Dipstick	*S. haematobium*	4.1 (6/146)	-			-	-	-			
**Democratic Republic of the Congo**	KK	*S. mansoni*	82.7 (277/335)	-	Epidemiology/parasitology	8–16	43.2	32	24.7	2011	2014	[196]
	KK	*S. mansoni*	8.9 (47/526)	-	Cross-sectional survey	7–13	8.8	-	-	2016	2017	[197]
	Urine microscopy	*S. haematobium*	0 (0/526)	-			-	-	-			
	Urine microscopy	*S. haematobium*	17.4 (64/367)	-	Cross-sectional survey	>18	-	-	6.3	2016–2017	2019	[198]
	KK	*S. mansoni*	89.3 (176/197)	-	Cross-sectional survey	11–14	11.7	22.3	55.3	2011	2018	[199]
	KK	*S. mansoni*	57.8 (231/400)	-	Cross-sectional survey	9–14	18.6	28.6	52.8	2010	2016	[200]
	KK	*S. intercalatum*	48 (24/50)	-	Prevalence	9–15	50	20.8	29.2		2017	[201]
	KK	*S. intercalatum*	3.6 (6/167)	-	Epidemiological/parasitological survey	8–18	-	-	-	1994	1997	[202]
**Egypt**	KK	*S. mansoni*	35.8 (355/993)	-	Cross-sectional survey	-	-	-	-	1994–1996	2020	[203]
	KK	*S. mansoni*	1.8 (2/110)	-	Prevalence	6–15	-	-	-	-	2016	[204]
	Formol-ether	*S. mansoni*	0.9 (1/110)	-			-	-	-	-		
	CCA	*S. mansoni*	11.4 (4/110)	-			-	-	-	-		
**Equitorial Guinea**	KK	*S. intercalatum*	31.9 (114/357)	-	Evaluation	15–24 *	-	-	4.7	1988	1991	[205]
	KK	*S. intercalatum*	9.6 (27/281)	-			-	-	0.7	1989		
	KK	*S. intercalatum*	6.6 (23/345)	-			-	-	0.2	1990		
	KK	*S. intercalatum*	13 (39/305)	-	Cross-sectional survey	0–24	-	-	9	1990	1993	[206]
**Ethiopia**	KK	*S. mansoni*	42.9 (136/317)	-	Cross-sectional survey	6–15	20.5	10.7	11.7	2017	2019	[207]
	KK	*S. mansoni*	76.3 (293/384)	-		5–19	21.6	29.4	25.5	2013	2014	[208]
	KK	*S. mansoni*	24 (120/500)	-		6–18	70	30	20	2014	2016	[209]
	KK	*S. mansoni*	58.6 (295/503)	-		5–19	34.2	35.5	30	2015	2017	[210]
**Gabon**	Urine microscopy	*S. haematobium*	77.7 (66/85)	-	Evaluation	6–39	-	-	34.8	-	2014	[211]
	qPCR	*S. haematobium*	98.5 (65/66)	-			-	-	-			
	Urine microscopy	*S. haematobium*	39.9 (103/258)	-	Longitudinal	6–30	-	-	-	2016–2018	2019	[212]
**Gambia**	POC-CCA	*S. haematobium*	23.3 (456/1954)	47.98/79.44	Prevalence	7–14	-	-	-	2015	2017	[213]
	Dipstick	*S. haematobium*	17.1 (334/1954)	47.01/81.54			-	-	-			
	Urine microscopy	*S. haematobium*	10.1 (198/1954)	-			-	-	2.7			
	KK	*S. mansoni*	0.3 (5/1954)	60/76.76			-	-	0			
**Ghana**	qPCR	*S. haematobium*	48.5 (79/163)	100/59.2	Prevalence						2020	[214]
		*S. mansoni*	28.7 (94/328)	-	Epidemiology/Prevalence	7–17	50	35.1	11.7	2017		
		*S. mansoni*	70.1 (54/77)	-	Longitudinal	0–4	-	-	-	2018	2020	[215]
			7.9 (9/108)									
			13.7 (13/96)									
		*S. mansoni*	80.1 (153/191)	-		5–16	-	-	-			
			39.9 (89/224)									
			35.9 (86/240)									
		*S. mansoni*	79.1 (200/253)	-		>17	-	-	-			
			32.1 (107/332)									
			34.8 (100/286)									
	Urine microscopy	*S. haematobium*	5.2 (4/76)	-		0–4	-	-	-			
			0 (0/105)									
			13.2 (11/87)									
		*S. haematobium*	23.8 (59/249)	-		5–16	-	-	-			
			5.8 (14/236)									
			27.6 (63/230)									
		*S. haematobium*	10.3 (32/308)	-		>17	-	-	-			
			2.9 (10/346)									
			15.1 (41/272)									
**Guinea**	KK	*S. mansoni*	66.2 (278/420)	-	Cross-sectional survey	9–14	8.8	24	33.3	-	2011	[216]
	Urine microscopy	*S. haematobium*	21.0 (88/420)	-			12.1	-	8.8			
**Guinea-Bissau**	Urine microscopy	*S. haematobium*	20 (18/90)	-	Prevalence	6–15	-	-	-	2011	2016	[217]
	Haematuria	*S. haematobium*	61.1 (11/18)	-								
**Kenya**	Urine microscopy	*S. haematobium*	83.3 (95/114)	-	Evaluation	6–15	-	-	-	1996–2010	2014	[218]
	Hematuria	*S. haematobium*	86.0 (98/114)	-			-	-	-			
	cSEA-ELISA	*S. haematobium*	79.8 (91/114)	-			-	-	-			
	PCR	*S. haematobium*	100 (114/114)	-			-	-	-			
	KK	*S. mansoni*	93.9 (1731/1844)	-	Evaluation	8–12	10.2	46.9	42.9		2015	[219]
	KK	*S. mansoni*	60.5 (2458/4064)	-	Prevalence	5–19	49	35.8	15.2	2012	2012	[220]
**Liberia**	KK	*S. mansoni*	87 (333/384)	-	Prevalence	1–>40	25.3	29.2	31.8	1980	1985	[221]
	Urine microscopy	*S. haematobium*	42 (177/423)	-								
	KK	*S. mansoni*	78 (276/353)	-	Prevalence	-	-	-	-	-	2018	[222]
**Madagascar**	Urine microscopy	*S. haematobium*	100 (79/79)	100/100	Prevalence	15–33	-	-	-	2010	2020	[223]
	qPCR		81 (64/79)	-			-	-	-			
	KK	*S. mansoni*	5 (97/1934)	-	Baseline sentinel study	7–10	-	-	0.9	2015	2016	[224]
	Urine microscopy	*S. haematobium*	30.5 (594/1946)	-			-	-	15.1			
	KK	*S. mansoni*	73.6 (215/292)	-	Prevalence	5–14	36.7	31.2	32.1	2015	2017	[225]
**Malawi**	Urine microscopy	*S. haematobium*	13 (18/143)	-	Cross-sectional survey	0.6–6	58	33	9	2012	2016	[226]
	Urine microscopy	*S. haematobium*	12.5 (50/400)		Prevalence	7–12	8.25	1.75	2.5	2012	2017	[227]
**Mali**	Urine microscopy	*S. haematobium*	51.2 (173/338)	-	Prevalence	1–4	35.5	-	15.7		2011	[228]
	Urine microscopy	*S. haematobium*	88 (570/648)	-	Cross-sectional	7–14	-	-	48.8	2004	2012	[229]
	KK	*S. mansoni*	17.3 (112/648)	-			-	-	15.6			
	KK	*S. mansoni*	12.7 (81/640)	-			-	-	9.4	2010	2012	
	Urine microscopy	*S. haematobium*	61.7 (395/640)	-			-	-	13.8			
**Mauritania**	Urine microscopy	*S. haematobium*	4 (86/2162)	-	Cross-sectional survey	-	-	-	-	2014–2015	2017	[230]
	KK	*S. mansoni*	7.1 (92/1297)	-	Epidemiological survey	5–12	-	-	-	-	1997	[231]
	Urine microscopy	*S. haematobium*	15.6 (48/307)	-	Prevalence	7–17	-	-	-	-	2019	[232]
**Mozambique**	Urine microscopy	*S. haematobium*	60.4 (11492/19039)	-	Cross-sectional survey	5–55	-	-	17.7	-	2018	[233]
	Urine microscopy	*S. haematobium*	59.1 (600/1015)	-	Cross-sectional survey	5–12	-	-	-	2005–2007	2014	[234]
	Urine microscopy	*S. haematobium*	47 (39166/83331)	-	Prevalence	7–22	-	-	17.9	-	2009	[235]
	KK	*S. mansoni*	8.7 (7250/83331)	-			-	-	-			
**Namibia**	KK	*S. mansoni*	4.4 (913/17896)	-	Mapping	3–19	-	-	-	-	2015	[236]
	Dipstick	*S. haematobium*	5.0 (895/17896)	-			-	-	-			
	CCA	*S. mansoni*	4.4 (787/17896)	-			-	-	-			
	Urine microscopy	*S. haematobium*	5.1 (913/17896)	-			-	-	-			
**Niger**	Hematuria	*S. haematobium*	58.4 (52/89)	-	Evaluation	10–15	-	-	-	-	2011	[237]
	Urine microscopy	*S. haematobium*	49.4 (44/89)	-			-	-	-			
	PCR	*S. haematobium*	57.3 (51/89)	100/86			-	-	-			
**Nigeria**	Urine microscopy	*S. haematobium*	21.3% (26/122)	-	Comparative	31–55	-	-	-	-	2018	[238]
	Urine microscopy	*S. mansoni*	8.9 (49/551)	-	Cross-sectional survey	1–90	80.8	15.4	3.8	2013	2016	[237]
	Urine microscopy	*S. haematobium*	8.3 (46/551)	-	Cross-sectional survey		69.4	0	30.6			
	Urine microscopy	*S. intercalatum*	5.7 (98/1709)	-	Malacological survey	5–15	-	-	-	1987	1989	[184]
	Urine microscopy	*S. haematobium*	44.1 (64/145)	-	Cross-sectional survey	5–59	2	26	11	2017	2019	[239]
	Urine microscopy	*S. haematobium*	22.7 (163/718)	-	Cross-sectional survey	10–23	89.57	-	10.43	2015	2016	[187]
	Urine microscopy	*S. haematobium*	50.0 (220/443)	-	Cross-sectional survey	5–14	39.5	7	4.5	2003	2008	[240]
	Urine microscopy	*S. haematobium*	14.5 (55/380)	-	Cross-sectional survey	5–14	11.3	1.8	1.3	2011	2017	[241]
**Rwanda**	KK	*S. mansoni*	2.7 (82/3052)	-	Cross-sectional survey	-	-	-	-	2007	2008	[242]
**São Tomé and Príncipe**	KK	*S. intercalatum*	11 (332/3030)	-	Cross-sectional survey	5–15	54	38	8	1991	1994	[44]
**Senegal**	KK	*S. mansoni*	80 (70/88)	-	Evaluation	2–83	54.55	15.9	9.1	2006	2008	[243]
	Urine microscopy	*S. haematobium*	72 (63/88)	-			50	-	21.6			
	qPCR	*S. mansoni*	73 (64/88)	-			-	-	-			
	qPCR	*S. haematobium*	55 (48/88)	-			-	-	-			
**South Africa**	Urine microscopy	*S. haematobium*	19.8 (78/394)	-	Prevalence	16–23	-	-	-	2010–2012	2020	[223]
	qPCR		23.1 (91/394)	-			-	-	-			
	Urine microscopy	*S. haematobium*	1.0 (11/1143)	-	Cross-sectional survey	1–5	-	-	-	2018	2019	[244]
	KK	*S. mansoni*	0.9 (9/998)	-			-	-	-			
	Urine microscopy	*S. haematobium*	40.2 (169/380)	-	Prevalence	10–15	61	-	-	2014	2018	[245]
	Urine microscopy	*S. haematobium*	31.8% (225/708)	-	Cross-sectional survey	10–12	-	-	26.7	2009–2010	2014	[246]
	qPCR	*S. haematobium*	25.4 (180/708)	-			-	-	-			
	Urine microscopy	*S. haematobium*	37.5 (120/320)	-	Prevalence	10–15	-	-	-	2015	2017	[247]
**Sudan**	KK	*S. mansoni*	36 (1020/2832)	-	Cross-sectional	10–24	-	-	-	-	1993	[248]
	Urine microscopy	*S. haematobium*	38.9 (58/149)	-	Comparative	5- >20	-	-	2	2011–2013	2018	[249]
	ELISA	*S. haematobium*	81.2 (119/149)	-			-	-	-			
**Swaziland**	Urine microscopy	*S. haematobium*	5.3 (21/395)	-	Prevalence	6–12	-	-	-	2010	2011	[250]
	Urine microscopy	*S. haematobium*	6.1 (18/295)	-		<5- >19	-	-	-	-	2010	[251]
**Tanzania**	KK	*S. mansoni*	85.2 (253/297)	89.7/72.8	Cross-sectional survey	7–16	30.6	39.1	15.5	2015	2018	[252]
	qPCR		92.9 (276/297)	98.7/81.2			-	-	-			
	POC_CCA		94.9 (282/297)	99.5/63.4			-	-	-			
	KK	*S. mansoni*	68.9 (641/930)	-	Cross-sectional survey	1–95	55.2	20.4	12.9	2016	2019	[253]
	POC_CCA	*S. mansoni*	94.5 (878/929)	-			-	-	-			
	KK	*S. mansoni*	90.6 (752/830)	-	Cross-sectional survey	5–19	24.1	38.4	28.1	2017	2020	[254]
	KK	*S. mansoni*	15.1 (898/5952)	-	Cross-sectional survey	7–16	-	-	-	-	2015	[255]
	Urine microscopy	*S. haematobium*	8.9 (519/5952)	-			-	-	-	-		
	KK	*S. mansoni*	84.01 (431/513)	-	Cross-sectional survey	6–16	34.11	39.91	25.99	-	2016	[256]
	Urine microscopy	*S. haematobium*	11.6 (13/112)	-	Prevalence		-	-	-	2009–2010	2020	[223]
	qPCR	*S. haematobium*	19.6 (22/112)	-			-	-	-	-		
	KK	*S. mansoni*	1.3 (4/310)	-			-	-	-	-		
**Uganda**	Urine-CCA Dipstick	*S. mansoni*	56.7 (146/258)	99.1/89.3	Surveillance	5–10	-	-	-	-	2018	[257]
	SEA ELISA	*S. mansoni*	75.1 (193/258)	97.7/49.5			-	-	-	-		
	KK	*S. mansoni*	39.3 (1203/3058)	-	Prevalence	1–5	60.7	21.8	17.5	2012–2013	2015	[258]
	KK	*S. mansoni*	40.8 (1850/4534)	-	Prevalence	10–14	-	-	-	2009–2010	2011	[259]
	KK	*S. mansoni*	27.2 (352/1295)	-	Prevalence	0.4–6.5	18.7	6	2.5	2009	2010	[260]
	ELISA	*S. mansoni*	66 (38/58)	-			-	-	-			
	KK	*S. mansoni*	47.6 (342/719)	-		15–70	29.2	12.7	5.7			
	ELISA	*S. mansoni*	41.0 (34/83)	-			-	-	-			
	Urine microscopy	*S. haematobium*	2.51 (24/955)	-		5–17	-	-	-	2007–2011	2018	[261]
**Zambia**	Urine microscopy	*S. haematobium*	61 (90/147)	-	Evaluation	7–14	26	-	19	-	2020	[262]
	KK	*S. mansoni*	0.01 (2/147)	-			-	-	-			
	DDIA	*S. haematobium*	51 (75/146)	60/61			-	-	-			
	IHA		56 (82/146)	74/72			-	-	-			
	Urine microscopy	*S. haematobium*	20.7 (328/1583)	-	Prevalence	5–17	-	-	-	2007	2010	[263]
	Urine microscopy	*S. haematobium*	28.6 (279/975)	-	Prevalence	9–16	84.9	-	15.1	(2007–2015)	2018	[264]
	Urine microscopy	*S. haematobium*	31.5 (494/1570)	-	Prevalence	9–15	75.5	-	24.3	2011–2015		
	KK	*S. mansoni*	42.4 (304/719)	-	Cross-sectional survey	7–50	61.2	26	12.8	-	2014	[265]
**Zimbabwe**	KK	*S. mansoni*	11.0 (10/91)	-	Comparative	1–12	2.1	8.8	-	2012	2014	[266]
	Urine microscopy	*S. haematobium*	52.8 (48/91)	-			41.8	-	11			
	SmCTF-RDT	*Schistosoma spp*	83.5 (76/91)	-			-	-	-			
	Urine microscopy	*S. haematobium*	18.0 (2347/13037)	-	Cross-sectional survey	10–15	12.4	-	5.6	2010–2011	2014	[267]
	KK	*S. mansoni*	7.2 (882/12249)	-			3.6	1.4	0.3			
	Urine microscopy	*S. haematobium*	18.7 (61/325)	-	Cross-sectional survey	17–49	93.4	-	6.6	2016–2017	2019	[268]
	Urine microscopy	*S. haematobium*	13.3 (71/535)	-		<5	93	-	7			

Tanzania has the second highest prevalence of schistosomiasis in Africa with about 23 million people infected, and is co-endemic for *S. haematobium* and *S. mansoni* [147,269]. *S. hematobium* is more prevalent throughout the country, especially in coastal areas [269,270] while *S. mansoni* is more focal and widely distributed along the shores and islands of large water bodies, including Lake Victoria [269,271,272,273]. The entire Tanzanian population is at risk of infection due to the country-wide distribution of the disease.

The Democratic republic of Congo (DRC) in Central Africa and Ghana has the third highest number (15 million) cases of schistosomiasis in Africa [25]; *S. mansoni, S. haematobium* and *S. intercalatum are endemic* (Table 2). Despite a relative paucity of studies, the trend appears to have been a spread of schistosomiasis from formally endemic areas to new areas over the last 60 years [274,275]. The most recent reports on prevalence in the DRC come from two cross-sectional surveys performed in 2016–2017 [197,276].

Both *S. haematobium* and *S. mansoni* are prevalent in in Ghana [214,215,277]. A recent longitudinal study involving 2623 participants including pre-school aged children (pre-SAC), SAC and adults in Ghana recorded an overall prevalence of 44.2% for *S. mansoni* and 11.9% for *S. haematobium*, with SAC recording the highest *S. haematobium* prevalence [215]. The adult participants had a significantly higher prevalence of *S. mansoni* across the three sampling sites due to local economic activities and differences in snail habitat in the communities increasing the risk of infection [215] (Table 2).

In North Africa and the Middle East, it is estimated that 12.7 million individuals have schistosomiasis [11,278]. Egypt previously had the highest number of recorded schistosomiasis cases in northern Africa for a long period but has recorded reduced prevalence in recent years due to successful implementation of control and elimination initiatives [279]. Morocco and Tunisia have successfully achieved transmission interruption of the disease (Figure 2) [280,281,282,283]. This came about through effective chemotherapy treatment programs, as well as targeted snail control programs and the use of molluscicides [281]. Elsewhere in North Africa, Algeria still awaits WHO confirmation of schistosomiasis interruption, as noted in the 2001–2011 WHO progress report, citing no new cases for ten years [284]. However, in 2003, it was estimated that 23.4% of the Algerian population was at risk of infection and 2.4 million were estimated to be infected [11]. There are no recent reports on schistosomiasis in Algeria and it is unclear if surveillance for the disease is being carried out; thus its true status remains unclear.

In East and Southeast Africa, *S. mansoni* is the dominant species present (Figure 2, Table 2). *S. mansoni* was first reported in Uganda in 1902 and high incidences of *S. mansoni* infection were reported in 1924; infection rates remained high in 1958 [285,286,287,288]. In 1949, an extraordinary *S. mansoni* prevalence of 95.2% was recorded in men around Lake Mobutu in Uganda, with all fishermen tested found to be infected [289]. *S. haematobium* and *S. mansoni* are both highly prevalent in Uganda while *S. intercalatam* was reported in some areas in 1978 [290] (Table 2). Recent reports of schistosomiasis in Uganda have focussed primarily on *S. mansoni* due to its increasing country prevalence, while *S. haematobium* has a more localised sporadic occurrence [285]. The prevalence of *S. mansoni* in Ethiopia remains high with recent surveys reporting prevalence levels between 24-76.3% determined using the KK procedure (Table 2). Similar to Tanzania, Mozambique is endemic for both *S. mansoni* and *S. haematobium*, however the prevalence of *S. haematobium* is higher in Mozambique, whereas *S. mansoni* is more prevalent in Tanzania (Table 2).

The prevalence of schistosomiasis in South Africa is generally high but low among pre-SAC (Table 2). A 2018 survey in KwaZulu-Natal recorded a prevalence of 1% for *S. haematobium* and 0.9% for *S. mansoni* among pre-SAC [291], while a cross-sectional study in 2017 determined 37.5% prevalence among SAC (Table 2). Another study in 2014 recorded a high prevalence of 31.8% of *S. haematobium* prevalence among girls aged 10-12 years by urine microscopy [246], while a report published in 2020 recorded a slightly lower prevalence of 19.8% for *S. haematobium* using urine microscopy and 23.1% by qPCR among young women aged 16 to 23 years [223] (Table 2). Swaziland and Zambia have high prevalence of *S. haematobium,* while both *S. haematobium* and *S. mansoni* are endemic in Zimbabwe, Namibia and Malawi (Table 2).

Another hindrance to control is the crucial lack, in a number of sub-Saharan countries, of reliable and accurate data on the prevalence, intensity of infection, epidemiology, and geographic spread of schistosomiasis. This is partly due to inadequate or a complete lack of surveillance and monitoring of the disease which leads to an inability to predict transmission areas and identify vulnerable populations [291,292,293]. Whereas national control programs have been put in place in many African nations, led by the Schistosomiasis Control Initiative [294], political instability, complacency and inadequate funding for control activities have led to re-emergence and even an increase in infection rates in some areas [27,285,295]. Nevertheless, transmission has been successfully interrupted in some regions of Africa who await verification from WHO confirming elimination (Figure 1) [154,280,281,284].

One of the real success stories in Africa is Morocco, which recorded transmission interruption in 2004, with no new cases in Moroccan natives recorded since [154,280,281]. In Burkina Faso, there has been a decline in the prevalence of schistosomiasis in SAC as a result of a decade of PC with PZQ [193,284]. A 2016 study to assess the impact of a decade of biennial MDA, and four and five rounds of PZQ per year in hyperendemic zones, showed that Burkina Faso had greatly reduced the prevalence and intensity of *S. haematobium* [193,284]. Indeed, based on WHO criteria, Burkina Faso has eliminated urogenital schistosomiasis as a public health concern in 8 out of its 13 regions [193,284]. Despite this success, prevalence still remains high in some areas, with 34.38% of surveyed SAC in the Central-East infected with *S. haematobium* and 20.94% in Sahel [193]. *S. mansoni* was identified in only two regions, with most infections occurring in Central-East region [193].

### 3.2. Stigmatisation Associated with Schistosomiasis, Particularly in Women Is still a Crucial Issue in Africa

The majority of schistosomiasis studies indicate no significant differences between the disease in males and females. However, it has been suggested that estimates of disease burden of schistosomiasis in females has been underreported [296,297,298]. One of the challenges still faced in Africa is gender disparity. Illiteracy, lack of education and lower economic and social status are still more common among African women than men [296,298,299,300]. Men are most often exposed to infection due to their occupations as farmers and fishermen. However, women also come in contact with cercariae regularly during washing or when they fetch water for domestic use in areas that lack a potable water supply. This frequent contact with water leaves them at risk of heavy infections that could cause permanent damage. The social consequences of schistosomiasis are also more severe in women than in men. For instance, haematuria resulting from *S. haematobium* infection, is still considered a sign of ‘coming of age’ in certain regions of Africa [20,21]. Some classify haematuria as a sign of sexually transmitted infection, while others consider it as a curse or witchcraft [20]. This perception is highly influenced by the level of education and local cultural beliefs [301]. A study among women reported that the presence of haematuria in urine is considered as part of menstruation, and is hence not regarded as a disease [301]. Several communities in Africa still view diagnosis of schistosomiasis in women as inappropriate, as they may be required to be subjected to physical examination or provide urine or stool samples [298]. In general, discussions about the influence of gender on NTD control programmes and interventions are limited, and if the issue of gender disparity and early intervention around diagnosis are not addressed, the true prevalence of schistosomiasis in women will remain uncertain, leading to issues governing treatment and control options in many communities [302,303].

### 3.3. Hybrid Schistosomes

Hybridization and introgression events have been identified in a range of organisms including schistosomes [8,304,305,306,307,308]. Acquisition of new genes due to hybridization can lead to development of new phenotypes that can change the pathology, virulence, hosts, and resistance of the parasite [306]. Schistosomiasis is a highly focal disease due to requirements of specific intermediate molluscan and definitive hosts, restricting schistosome species to ecological niches which restricts hybridization events. However, both natural and man-made changes have the ability to breakdown ecological hurdles (see section below, 4.0 Factors that determine the distribution of schistosomiasis in Africa), thereby introducing these parasite species and strains to new areas which can result in novel interactions between the hosts and parasites leading to hybridization events.

Among the human schistosome species, hybridization has been most commonly observed between *S. haematobium* and *S. intercalatum.* Prior to 1969, *S. intercalatum* was the only schistosome species present in Loum, a small town in Cameroon [307]. However, after increased haematuria in the population, a symptom usually associated with *S. haematobium*, it was discovered that introgressive hybridization had occurred, replacing *S. intercalatum* with *S. haematobium* [307]. This is due to the higher competitive fitness of male *S. haematobium* worms to mate with female *S. intercalatum* worms [9,309].

Additionally, it has been reported that the two strains of *S. intercalatum* found in Africa (Cameroon and DRC), exhibit different characteristics [310,311,312,313,314,315]. Kane and colleagues [315] carried out phylogenetic analysis utilizing the cytochrome oxidase subunit 1 (*cox1*), NADH dehydrogenase subunit 6 (*nad6*) and the small ribosomal RNA gene (*rrnS*) from the mitochondrial DNA of the two recognised strains of *S. intercalatum* and proposed that the strains should be treated as distinct taxa, with one of the strains, the DRC strain closer to *S. haematobium* and the West African strain aligning closer to the animal schistosomes *S. bovis* and *S. curassoni*. Phylogenetic information from this report was similar to an earlier study where random amplified polymorphic DNA (RAPD) markers were used to examine both *S. intercalatum* strains, which also reported significant differences between the two [316].

Molecular sequencing of eggs isolated from human stools and urine in northern Senegal revealed 15% of the eggs had hybrid genotypes, with fragments partially identical to *S. haematobium* and *S. bovis* [306]. Another Senegalian study, involving three closely related species: human *S. haematobium* and ruminant *S. bovis* and *S. curassoni* revealed the presence of *S. haematobium*/*S. curassoni* and *S. haematobium*/*S. bovis* hybrids in nearly 90% of the children surveyed; no *S. haematobium* hybrids were observed in ruminants, although *S. bovis*/*S. curassoni* hybrids were identified in these animals [8]. The hybridisation events were hypothesized to have occurred in this area due to ecological and climate changes leading to overlapping areas of distribution of these schistosome species and their intermediate and definitive hosts [8]. A study in two villages in the Republic of Benin also reported an introgression of *S. guineensis* genes in *S. haematobium* [317].

Hybridization and introgression events can influence the epidemiology of schistosomiasis in a number of ways, including the potential for new zoonotic forms to emerge, with important implications for control, as well as changes in pathogenicity, drug sensitivity, and parasite transmission.

### 3.4. Control Measures in addition to MDA Utilised in Africa

#### 3.4.1. Mapping Studies and Snail Control

Mapping and geospatial analysis of schistosomiasis is important to monitor transmission trends and prevalence, leading to better understanding of the disease burden and risk factors for infection, which will in turn result in more targeted control efforts and improved surveillance procedures. Mapping studies have gradually increased in Africa [201,259,318,319,320,321,322]. Communities that have performed mapping have been able to effectively scale up treatment and monitor applied control strategies [21]. Most reports focus on SAC, while a small number carried out on pre-SAC and adults indicate that these age groups are also at high risk of infection as they are constantly exposed to water bodies infested with cercariae [323,324]. Prevalence-based studies that exclude pre-SAC and adults can lead to reduced treatment levels during MDA, thus making these groups potential reservoirs of infection, leading to ongoing transmission risk to all ages.

It is now generally recognised that integrated strategies for schistosomiasis control and multisectoral approaches will be required to achieve elimination and these strategies have been increasingly advanced over the years [325,326,327,328]. Accurate data on parasite distribution, prevalence, at risk populations and other vulnerable groups including pre-SAC children will play important roles in the design of effective control measures. However, disease surveillance in low socio-economic endemic areas is greatly affected by low quality disease data as a result of underestimating infection rates due to inadequate health infrastructure, low parasite burden and reduced symptoms resulting from MDA and PZQ treatment. Predictive modelling can provide relevant information to aid in designing action plans for integrated and targeted control of schistosomiasis in many endemic areas.

Environmental factors including vegetation, temperature, elevation and precipitation are useful predictors of snail habitats which may be more reliable in assessing the distribution of schistosomiasis compared with prevalence survey data [158,329,330]. Environmental data can be obtained using remote sensing techniques (RS) and the geographic information system (GIS) [319,329,331,332]. A study in Ghana [333] utilised environmental variables including land surface temperatures (LST), normalized difference vegetation index (NDVI), and accumulated precipitation (AP) to analyse disease patterns with wide geographic coverage and varying levels of spatial and temporal aggregation. This assessment reported reduced rates of schistosomiasis, seasonal patterns across time zones and the associated schistosome infection rates, features that are useful in determining disease patterns [333].

Snail control plays a vital role in the interruption of schistosomiasis transmission [33,334,335,336]. Snail control using chemical molluscicides was introduced and used extensively in the 1950s-1970s in Africa, Asia and South America, until the inception of oral chemotherapeutics for humans led to its decline [32,337,338,339]. The most commonly used molluscicide is niclosamide, which is effective against all stages of the snail life cycle [32]. It has been shown to be effective in control and elimination programs for schistosomiasis in several African countries including Morocco [147]. The downside of niclosamide is that it can be toxic to the environment and other animals, expensive, labour intensive and does not prevent repopulation of snails after treatment [34,35,147,340]. Hence, an integrated control approach involving mass chemotherapy, snail control, improved access to portable water and sanitation, environmental modification, behaviourial changes and health education have been found to be more effective in schistosomiasis control that can lead to elimination [337,341,342]. This approach has led to transmission interruption and significant decline in disease prevalence in previously endemic areas in China. In Sub-Saharan Africa, the Zanzibar Elimination of Schistosomiasis Transmission (ZEST) project carried out betwen 2011–2017 using a combination of MDA, snail control (niclosamide) and behaviourial change measures, reported a significant decline in the prevalence of schistosomiasis and successfully eliminated the disease as a public health concern in most of its study sites [343]. It is worthy to note however, that transmission was not interrupted in these sites [343]. Another, a community-wide MDA and snail control program in rural Kenya reported significant declines in prevalence and intensity of the disease [344]. Alternative forms of snail control including introducing competitor snail or prawn species to feed on snail host populations should also be considered [150].

#### 3.4.2. Education and Knowledge

As with most NTDs, poverty, lack of adequate infrastructure and a low level of education are important factors affecting the prevalence of schistosomiasis in Africa. The WHO has highlighted education as part of its strategic plan for schistosomiasis control in Africa along with MDA and WASH activities [98]. The lack of knowledge about schistosomiasis and how it is transmitted in endemic regions is a major risk factor and can prevent successful implementation of control programs, including MDA, in some areas. A qualitative formative study targeted at behaviourial change intervention in Zanzibar reported that people’s knowledge of schistosomiasis symptoms, transmission and prevention was poor [345], despite previous control initiatives having been undertaken [346,347,348]. In Malawi during a MDA program in schools, children refused treatment as they were suspicious of the treatment with some scared of the large size or smell of the PZQ tablet, and because classmates felt dizzy after taking the drug [93]. There was also distrust in the community with some believing that the drug was a contraceptive, or they did not trust the medication in general. Others kept children home on treatment days as they considered schistosomiasis to be normal and could not understand the need for treatment [93]. After community education and involving the community in the MDA program, compliance went up, highlighting the need for community engagement in any control program. This suggests that regular school-based health education can play an important role on health behavior, and can aid in reducing infection prevalence among SAC [298].

An interactive board game, Schisto and Ladders™, which included information on the mode of schistosome transmission, behavioural risks associated with transmission, symptoms of infection and information on what to do to seek treatment, prevention of reinfection and control strategies for schistosomiasis was developed and tested in schools in an endemic area (near Abeokuta), in Nigeria [349]. By the end of the study, increase in knowledge of schistosomiasis and behavioural changes were observed among the participants [349]; as diagnostics were not included in the study, the impact of the intervention on disease prevalence in the area is unknown. In addition to increasing compliance in MDA [350], increased knowledge on transmission will help inform the community on how to prevent re-infection–an important aspect considering the current lack of a schistosomiasis vaccine.

A systematic review to understand knowledge, attitudes and practices (KAP) about schistosomiasis in a number of communities in sub-Saharan Africa identified several short comings [291]. The survey revealed there was exclusion of children under five, and therefore their caregivers, in education programs and there was a focus on SAC to the exclusion of other community members. This is generally due to greater ease in implementing education programs in schools and the belief that what children learn in school is reported back to and taken on board by their caregivers; however, this excludes a large portion of the community who do not have children. The review also showed gender disparity, with males more likely to have higher levels of knowledge of schistosomiasis, more likely to be exposed during fishing and farming and, possibly, more likely to be targeted by education programs, or having gained knowledge of the disease from other fishermen and farmers in the area.

Misconceptions are also a potential issue either from improperly understood education programs, or due to superstitious beliefs already present in the community [18]. Schistosomiasis as a sexually transmitted disease is a common misconception that can be a barrier to seeking treatment, particularly in women due to the associated stigma [18]. A study in Uganda reported that 78% of a study population of 370 SAC were aware of schistosomiasis but did not have an idea of its mode of transmission [351]. Most infections were likely to have come from lakes in the area and community-based education may have led to avoidance of high transmission areas, however lakes are the only water source available for the majority of the community. It is therefore important to pair education with health and hygiene infrastructure to provide options and access to safe drinking water, and the provision of toilets to increase sanitation and decrease environmental contamination with eggs. Several studies report that successful prevention programs among vulnerable groups, especially children, women and individuals whose occupations require contact with freshwater bodies could reduce the rate of transmission of schistosomiasis; and could eventually lead to reduced prevalence of the disease in the general population [326,349,352].

## 4. Factors That Determine the Distribution of Schistosomiasis in Africa

As emphasized, schistosomiasis is a focal disease that requires the presence of the intermediate snail hosts. Various factors including climate change, flooding and manmade structures, particularly dams, can affect the habitats and spread of the infection by expanding snail habitats and increasing transmission these sites as well as introducing them to new locations [3].

### 4.1. Climate Change

Climatic conditions including temperature, precipitation and altitude are major factors that affect the distribution of *Schistosoma* species and are important to understand for control programs [353,354,355]. It has been reported that *S. mansoni* infections may increase in parts of eastern Africa as temperatures continue to rise, as it will influence the growth, survival and distribution of the parasite and its intermediate molluscan hosts [146,356,357,358]. It has also been noted, however, that infections may decrease in certain areas as increases in temperature may result in unfavourable conditions for both schistosomes and their snail hosts [146,356,357,358]. Pedersen et al. [359] reported a downward trend in schistosomiasis infections in Zimbabwe over three decades due to a warmer and drier climate, although MDA and behavioural changes may have also played a role in this downward trend [359,360]. A study in Ethiopia suggested that altitudes lower than 800 m above sea level favour the transmission of *S. haematobium* while altitudes between 1300 and 2000 m above sea level favour the transmission of *S. mansoni* [361]. An epidemiological survey of intestinal schistosomiasis in SAC living around the high altitude crater in Western Uganda reported that the geographic distribution of schistosomiasis may be altered due to increasing temperatures in tropical and sub-tropical regions [351]. The study suggested that altitudes above 1400 m, which previously have shown no possibility of schistosomiasis transmission due to being too cold, were now experiencing transmission of the disease due to the increasing temperature resulting from climate change [351,362].

### 4.2. Artificial (Man-Made) Activities

Dams are crucial infrastructure in many ways including maintaining water availability and hydroelectric power production. However, construction of dams can lead to ecosystem imbalance, including negatively affecting the life cycle of natural schistosome intermediate snail host predators, such as *Macrobrachium* spp. river prawns, thereby leading to a rapid spread of schistosomiasis [363,364,365]. One example is the construction of the Diama Dam in Senegal that led to the introduction and spread of schistosomiasis in nearby villages [366,367]. Reproduction of river prawns, which normally fed on schistosome intermediate snail hosts, was affected as female prawns were prevented from migrating downstream while juvenile prawns were prevented from migrating upstream to complete their life cycle [150,366,367]. Restoring these river prawns to the area led to a decrease in schistosomiasis [150]. A study in Raffierkro, Côte d’Ivoire, during the construction and initial operation of a dam, showed low schistosomiasis levels but reported an increase in uninfected snail hosts as a result of the construction [368]. To date, however, there has been no follow-up report of the current status of the intermediate snail hosts and the level of schistosome transmission in this locality.

### 4.3. Human Migration

Several studies have reported an increased risk of schistosomiasis as a result of infected migrants moving from endemic regions. Forced displacement, rural-urban migration and refugee settlements are good examples of such migration. Studies among refugees and asylum seekers from sub-Saharan Africa reported a high number of urogenital schistosomiasis cases among migrant groups, which were mostly from West Africa [141,369]. Infected migrants most likely acquired the infection during childhood while resident in Africa [141,369]. Schistosomiasis has also been reported in seasonal migratory of workers who travel from uninfected areas to and from endemic areas, have frequent water contact and are faced with poor sanitary conditions [370]. Schistosomiasis is also diagnosed in returned travelers in non-endemic areas who visit endemic countries in Africa [371,372,373,374].

As referred to earlier, there have been some recently reported cases of *S. haematobium* in parts of Europe, including autochthonous infections in Corsica, France, likely introduced due to migration of African refuges [5,6,369,375,376,377]. Sustained transmission then occurred due to the presence of susceptible intermediate snail hosts. A case study from Spain identified *S. haematobium* in a patient originally from the Dominican Republic; the subject had no history of travel to schistosome-endemic areas, and was diagnosed based on the presence of haematuria, egg morphology, IHA and PCR [378]. The patient had experienced chronic symptoms since childhood, placing the likely source of infection as the Dominican Republic, where Lebanese immigrants may have inadvertently brought the infection in the 1980’s, indicating a potential infection lasting 30 years. However, *Bulinus* snails are known to be present in the South of Portugal, where the patient had previously travelled, and *Planorbarius*
*metidjensis* occurs in Spain (which can transmit *S*. *bovis*) where the patient had swum in local watercourses, which may point to more recent infection within Europe. Another recent study identified six cases of *S. haematobium* in African migrants in the Canary Islands [369]. These infections were likely acquired in their endemic communities before migrating to Europe, although show a reasonable cause for transmission to non-endemic areas where viable intermediate hosts are present, such as has occurred in France [369,375]. Schistosomiasis has also been identified in long-term residents of Barcelona who migrated from endemic countries in sub-Saharan Africa. In 2002–2016, a study in Barcelona enrolled individuals into a study based on previous travel to schistosomiasis endemic countries and tested for schistosome infection by microscopy (urine and faecal), antibody detection, and/or radiological suspicion [379]. Schistosomiasis is often overlooked in initial health screening of refugees and immigrants from schistosomiasis-endemic areas, particularly as symptomology is often low in chronic infections, increasing time for diagnosis. In the Barcelona study, sixty-one individuals were identified infected with *Schistosoma* spp. and 90% of the study participants displayed symptoms of schistosomiasis including eosinophilia, haematuria and abdominal pain [379].

## 5. COVID-19

The world is currently plagued by the COVID-19 pandemic, one of the most serious health crisis in recent times. As of 1 April 2020, the WHO guidance to member states recommended postponement of community based surveys, active case-finding activities, and mass drug treatment (MDA) campaigns in NTD programmes in order to comply with public health measures for prevention of COVID-19 treatment [380]. However, the WHO did recommend continued support for patients presenting to health care facilities and continuance of essential vector control measures where possible [380]. It has been documented that individuals in many countries are reluctant to present to health care facilities for fear of contacting COVID-19, or to avoid adding to the workload of already stressed hospitals, and this is likely to be true in NTD endemic regions [381,382,383,384]. Furthermore, many health care personnel formerly engaged in NTD activities are likely to be now involved in COVID-19 responses, leaving aspects of NTD control and treatment nonoperational.

The COVID-19 pandemic incidence was initially lower than expected in Africa and in many other developing nations, but, towards the end of 2020, there was a rapid increase in the number of cases. Cases have continued to increase, especially in schistosomiasis and other NTD endemic areas [385]. This has resulted in the suspension of many MDA programs, leading inevitably to future outbreaks of schistosomiasis [386] and other NTDs [387]. Additionally, it has been suggested that interactions between the severe acute respiratory syndrome coronavirus 2 (SARS-CoV-2) and parasitic infections may result in increased viral susceptibility and more severe outcomes in patients to COVID-19 in NTD-endemic regions [388].

## 6. Conclusions

Schistosomiasis continues to be a major public health issue in Africa, although the paucity of epidemiological studies in many African countries prevents full understanding of the disease in many settings and hinders control and elimination efforts. Effective health policies and surveillance, improved hygiene and sanitary conditions, as well as increased education and knowledge have led to the control and transmission interruption of the disease in some parts of Africa with assistance from government and commerical agencies.

The WHO has released a new 2021–2030 roadmap for NTDs. Among the recommendations, children should be the first target group for schistosomiasis intervention due to the adverse effects the disease has on their growth and development [174], and integration of health education that is community-wide, not just aimed at SAC, is essential. Additionally, early, accurate diagnosis and treatment in children reduces heavy infections and this will decrease the risk of severe disease and disability in juveniles [389]. Pre-SAC, women and adolescent girls are severely affected by the disease and there is a need to target and include them in public health interventions. More insight and understanding of religious, social, environmental and economic factors among children and adults alike, can assist in health intervention strategies directed at disadvantaged communities, as well as involving targeted communities in developing interventions. By involving these communities, uptake and longevity of programs can be increased. Having communities involved from early stages will also lead to ‘demystification’ around infection–something that may be particularly useful for alleviating stigma surrounding female infections, as well as around other cultural and societal norms that can prevent uptake of treatment.

Risk mapping is also emphasised as it is crucial to prioritise areas for targeted intervention and for surveillance after transmission interruption. Combined with environmental monitoring, risk mapping can give accurate predictions for outbreaks and modelling can incorporate flooding events which may increase risks of schistosomiasis infection as well as increasing snail dispersal. Countries in Africa are at many different stages of control; some are at very early stages while others are nearing transmission interruption. Therefore, different approaches will be required depending on status. Risk mapping may be the first step for many countries on the journey towards control and elimination of schistosomiasis, as it could allow for more targeted MDA in at-risk areas where geographical prevalence information may be lacking. On the other hand, environmental and snail monitoring may be of more use in countries approaching transmission interruption to determine risk levels and monitor for outbreaks or ‘rebounding’ infection.

Overall there is a need for current, up to date, information of the prevalence and geographical spread of schistosomiasis to help inform control programs. Multifaceted control that includes snail monitoring and control, health education, and increases in public health infrastructure combined with widespread chemotherapy, as well as poverty alleviation, will be critically important to reach the goal of eventually eliminating schistosomiasis from Africa.

## Figures and Tables

**Figure 1 tropicalmed-06-00109-f001:**
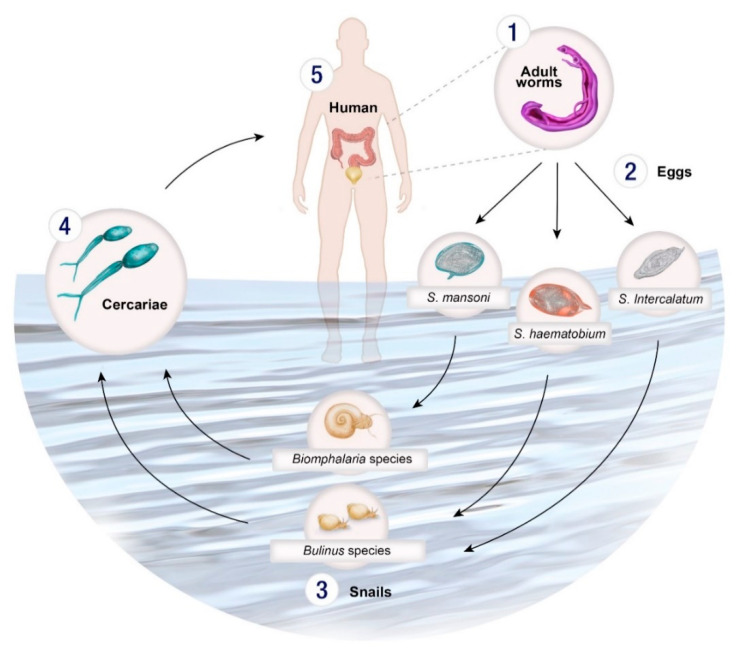
Life cycle of *Schistosoma* spp. (1) Male and female adult worms reproduce sexually in the venous system of the bladder (*S. haematobium*) or the bowel (*S. mansoni, S. intercalatum, S. guineensis*) producing eggs which are excreted in urine or via faeces, or are retained in body tissues, such as the liver. (2) The eggs hatch upon contact with water releasing miracidia which then penetrate a specific intermediate molluscan host. (3) Within the snail host, the miracidia develop into sporocysts and asexually reproduce daughter sporocysts which in turn produce cercariae. (4) The cercariae emerge from the snail and directly penetrate the skin of the (5) human host and transform into schistosomula. The schistosomula migrate via the circulatory system to the lungs and then the heart before arriving in the liver where they mature. Once mature the adult worms exit the liver and pair up to mate in the mesenteric vessels of the bowel bowel (*S. mansoni, S. intercalatum, S. guineensis*) or bladder (*S. haematobium*).

**Figure 2 tropicalmed-06-00109-f002:**
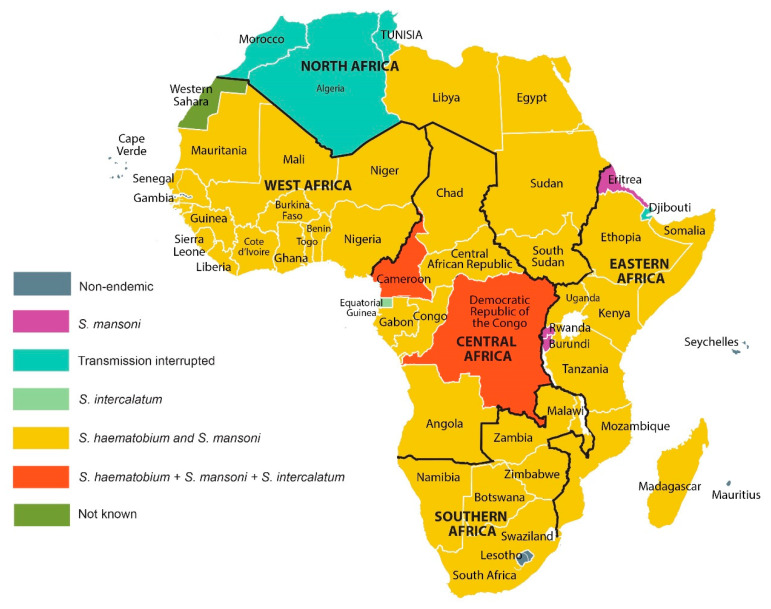
Distribution of schistosome infections in Africa modified from references [176,177].

**Table 1 tropicalmed-06-00109-t001:** African schistosome species and their geographical distribution.

	Species	Intermediate Host	Definitive Hosts	Site of Infection	Geographical Distribution	References
**Intestinal schistosomiasis**	*S. mansoni*	*Biomphalari* spp.	Humans, rodents	Intestinal mesenteric veins	Sub-Saharan Africa, Madagascar, the Middle East, the Caribbean, South America	[1,10]
	*S. intercalatum* and *S. guineensis*	*Bulinus* spp.	Humans, non-human primates (excluding apes)	Intestinal mesenteric veins	Central Africa, West Africa, Madagascar	[1,10,11]
**Urogenital schistosomiasis**	*S. haematobium*	*Bulinus* spp.	Humans, non-human primates (excluding apes)	Urogenital veins	Sub-Saharan Africa, the Middle East, Corsica (France)	[1,10]
**Animal intestinal schistosomiasis**	*S. mattheei*	*Bulinus* spp.	Cattle, sheep, goats	Intestinal mesenteric veins	Southeastern and Central Africa	[12,13,14,15]
	*S. curassoni*	*Bulinus* spp.	Cattle, sheep, goats	Intestinal mesenteric veins	West Africa	[12,13,14,15]
	*S. bovis*	*Bulinus* spp.	Cattle, goats, sheep, horses, camels, pigs	Intestinal mesenteric veins	North, East, West and Central Africa, the Middle East and Mediterranean (Europe) region	[12,13,14,15]

## Data Availability

Data sharing not applicable. No new data were created or analysed in this study. Data sharing is not applicable to this article.

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
