# Peer review of "Schistosomiasis with a Focus on Africa"

_tropicalmed, 2021, doi:10.3390/tropicalmed6030109_

Round 1

Reviewer 1 Report

General comments

Overall, the review from Aula et al. was a comprehensive and well-rounded piece in terms of the topics covered, and I commend the authors for combining so much literature into one paper. I particularly enjoyed the sections on stigmatisation and education/misconception which pointed me to some interesting literature. The review spans many aspects and discusses the facts of schistosomiasis in SSA, stemming from basic biology to control and future complications in elimination of the disease, with only some areas that need clearing up and tidying to help the readers follow the reviewed literature.

I do feel somewhat disappointed in the conclusions that there was a lack of insightful comments or in-depth analysis or recommendations for future schistosomiasis control and elimination in SSA, which is the obvious end goal for schistosomiasis, and was how I expected this piece to end. It would be nice to see more of the authors opinions and thoughts shine through in the conclusion after such a long and extensive discussion of the literature which presented the reader with many of the facts. What challenges pose themselves in SSA that do not in other endemic regions? How will these be overcome? What are the priorities for countries with ongoing control or those that are just starting? What about elimination surveillance when that is reached? These are just a few of the topics that could be expanded on here and could significantly improve the paper and make of use for the scientific community interested in schistosomiasis in SSA and make it a useful reference. The manuscript currently leaves the reader wanting to have a mor refined and thoughtful summary of what has been discussed (since there is so much preceding information). Adding such sections will also greatly improve the likelihood of this paper being referenced, rather than just pointing to relevant literature.

Some of the more descriptive text in the paper could be removed (such as the lengthy country by country facts and figures), and other parts can be combined (e.g. repeated mentions of MDA and clinical presentations), to make the manuscript more concise and follow the headings the authors have provided.

After addressing these general comments and the line-by-line comments below, the reviewer would be happy to see this paper published.

Line by line

8 – also a disease of animals / livestock

11 – estimated twice in sentence – 800 million what? State people / animals etc

16 – ‘a need’

16 – Neglecting fact that schistosomiasis a disease of animals in abstract – be explicit stating ‘human African schistosomiasis’

24 – 2.0 introduction? Or 1.0?

26 – tropical and sub-tropical

31 – Sub-Saharan Africa (use capital S). Change throughout.

31 – First use of Schistosoma in main text – spell out in full.

37 – Introduce animal schistosome species before discussing hybrids.

51 – Remove ‘for schistosomiasis’ – reads better.

52 – 53 – Discussing use of MDA seems repetitive from lines 39 / 40 in intro. Condense and refine.

55 – term ‘new patent infection’ does not read correctly in this instance since person is already infected, seems more as a reinfection. Re write sentence and also separate sentence away from part about re-infection which seems inappropriately tagged on the end here.

 57 – Maybe also state here what the approximate age range of SAC is

58 – what daily activities? State a couple

62 – Reference for age related immunity at end of this sentence

66 – as now discussing lifecycle, use ‘gastropod’ instead of ‘molluscan’ intermediate host

71 – 72 – sentence related to chronic disease not part of life cycle, move this to appropriate section

75 – 86 – This paragraph and one above doesn’t clearly state lifecycle as you want – would leave naïve reader confused. Also discussing control efforts (i.e. snail control) in this section which seems inappropriate. I suggest tidying up this whole section as doesn’t provide full lifecycle to reader.

81 – should be distribution of schistosomes – it is the parasite itself which is ‘transmitted’ and not necessarily the ‘disease’ which is a description of the illness.

91-93 – repeat of first section of intro stating a disease of poverty.

129 – No mention of increased transmission of HIV and other STD/STI through genital bleeding associated with S. haematobium (see below comment 137 - 139)

.

137 – 139 – Should this not be discussed when talking HIV above? Sections need cleaning up and focusing

155 – 156 – discussing fibrosis of tissue related to eggs again – but this discussed above in clinical presentation section. Maybe simplify this part if already mentioning.

167 – 169 – Should this not be in the MGS section

179 – using ‘Schistosoma’ here but S. in other sub headings. Make consistent. Also. Italics?

226 – Is there evidence for resistance too for oxamniquine?

230-235 – This paragraph makes it sound like ALL SAC in SSA receive treatment, but we know this is not true and only regions where drug availability through donations is. Make this clear and include briefly current figures / statistics on treatment coverage.

249 – as above, distribution of parasite

250 – and also where overwintering Schistosoma infections in snails may survive? Look at literature from Perpignan group for S. haematobium over winter intramolluscan survival.

275 – Urine microscopy preceded with urine filtration?

323 – In PCR based methods, you have not mentioned the xenomonitoring and detection of schistosome DNA from collected snails, see recent publications by Schols et al 2019 and Pennance et al 2020, also those by Abbasi 2017 and Kane 2013, amongst others.

353 – Manson typo

391 – sensitivity

433 – typo?

441 – already mentioned co-endemicity in Tanzania above

In general – is all the descriptive text in this section necessary when it is displayed in Table 2?

456 – typo recently

454 – Unusual paragraph – starting with hybrids that discussed later and then moving on to epidemiological mapping?

587 – I think snail control still used in Asia for targeting Oncomelania hosts – snail control lost mainly in Africa and South America due to the technical logistics and finances of effective snail control.

590 – See recent papers from Knopp et al. of the ZEST study in Zanzibar, published in Lancet and PLOS. Results here showed minimal additive effect of snail control when compared to MDA alone. Worth discussing here.

595 – And environmental modification for snail control?

623 – also see recent publication from Person et al. in Acta Tropica, and others related with ZESt study where behavioural interventions and school awareness targeted. Celone 2016 also.

654 – intermediate snail host

757 – typo

774 – eventually?

Table 1

Sometimes ‘B.’ sometimes Bulinus / Biomphalaria, be consistent and maybe use Bu. And Bi.

Many more Bulinus and Biomphalaria species involved in transmission of S.m. and S.h. group species than listed here… I suggest for simplicity you just leave as Biomphalaria spp. and Bulinus spp. in the table.

Sh and Sm also Madagascar distribution (considered separate from SSA usually)

Table 2 – Where is the table caption / legend??

Hopefully in Figure caption it gives the explanation for how dividing studies (i.e. evaluation vs. comparative vs. etc)

Typo in table =  2009? Listed in table country column. Check all

Also can it be ordered alphabetically or something easier for reader?

Have one column for ref which gives first author name and year together. Would be easier to interpret for reader.

I’ll take the authors word that all numbers are correct in table. Cannot cross ref and check all.

Figure 1 – OK I see now how this might relate to your lifecycle section above. However, that section still needs work.  

References –

Check all for formatting and unusual ones – e.g. ref 373 = Africa, W.R.O.f. ? and ref 374 also unuaul? Not sure how these have been brought about.

Reviewer 2 Report

This is a well written and well documented review of the current state of knowledge of schistosomiasis in Africa.  While there were at least 30 reviews of one type or another on schistosomiasis in 2021 alone (and well over 300 papers), this paper does a good job of organizing and commenting on some of the social aspects of schistosomiasis control in Africa.  My only real criticism is what appears to be table 2 (running on the pages labeled as from page 2/58 through 14/58 of the Tables and Figures area - page numbering is somewhat confusing at best) does not have any legend or title or something that makes it understandable as a stand alone table.  This table is referenced at least 13 times in the text for everything from prevalence, to distribution, to diagnostics.  This may be better accomplished with separate tables rather than one huge table.
